# Modified Composite Biodegradable Mulch for Crop Growth and Sustainable Agriculture

**DOI:** 10.3390/polym16091295

**Published:** 2024-05-05

**Authors:** Bo Guo, Liyan Zhu, Xiaochan He, Xiaojun Zhou, Boru Dong, Jialei Liu

**Affiliations:** 1Key Laboratory of Agricultural Film Pollution Prevention and Control, Ministry of Agriculture and Rural Affairs, Institute of Environment and Sustainable Development in Agriculture, Chinese Academy of Agricultural Sciences, Beijing 100081, China; guobo9015@126.com (B.G.); 13179171167@163.com (B.D.); 2College of Resources and Environment, Shanxi Agricultural University, Jinzhong 030801, China; 3Jinhua Academy of Agricultural Sciences, Jinhua 321017, China; pasxzly@163.com (L.Z.); 164607454@163.com (X.H.); zxj741130@163.com (X.Z.)

**Keywords:** PBAT/PLA-PPC-PTLA biodegradable film, water vapor barrier performance, mechanical property, green sustainable development, “white” plastic pollution

## Abstract

Using biodegradable films as a substitute for conventional polyolefin films has emerged as a crucial technology to combat agricultural white pollution. To address the shortcomings in the tensile strength, water vapor barrier properties, and degradation period of PBAT-based biodegradable films, this investigation aimed to create a composite film that could improve the diverse properties of PBAT films. To achieve this, a PBAT/PLA-PPC-PTLA ternary blend system was introduced in the study. The system effectively fused PBAT with PLA and PPC, as evidenced by electron microscopy tests showing no apparent defects on the surface and cross-section of the blended film. The developed ternary blend system resulted in a 58.62% improvement in tensile strength, a 70.33% enhancement in water vapor barrier properties, and a 30-day extension of the functional period compared to pure PBAT biodegradable films. Field experiments on corn crops demonstrated that the modified biodegradable film is more suitable for agricultural production, as it improved thermal insulation and moisture retention, leading to a 5.45% increase in corn yield, approaching the yield of traditional polyolefin films.

## 1. Introduction

Numerous countries extensively employ agricultural coverings to augment crop yield and quality. Mulching technology plays a beneficial role in thermal insulation, moisture retention, and the improvement of soil physicochemical properties while also enhancing the quantity and activity of soil microorganisms [1]. This technique has emerged as a significant agricultural productivity measure in the arid and semi-arid regions of northern China [2,3,4]. In 2019, the global total plastic film usage for greenhouses and mulching was estimated to reach 7.4 million tons, marking a 69% increase from 4.4 million tons in 2012 [5,6,7,8,9]. However, the majority of these plastic films are composed of polyethylene (PE), polypropylene (PP), polyvinyl chloride (PVC), and polystyrene (PS), materials that are highly resistant to natural degradation and may lead to livestock ingestion and mortality, as well as the environmental issue known as “white pollution” [10,11,12,13,14,15]. Therefore, the use of biodegradable mulch films is an effective approach to tackle the problem of residual film pollution in farmland [16,17,18,19]. Biodegradable mulch films are made from biodegradable polymers or films developed from renewable resources, such as polybutylene adipate-co-terephthalate (PBAT), polylactic acid (PLA), polypropylene carbonate (PPC), and polybutylene succinate (PBS) [20,21,22,23]. The production of biodegradable mulch films has also become a focal point in the field of sustainable agricultural practices.

In the field of biodegradable polymer materials, PBAT has been extensively used in producing biodegradable mulch films in recent years due to its thermodynamic properties and film-forming characteristics, which are similar to those of PE [24,25,26]. However, its relatively weak intermolecular interactions result in a high water vapor transmission rate of approximately 800 g/(m^2^·24 h) for such films, far exceeding the 30–50 g/(m^2^·24 h) of traditional PE films. The significant evaporation of water removes heat from the soil, thereby impacting the thermal insulation performance of these films [27,28,29,30]. Additionally, the absorption of sunlight by ester groups and benzene rings leads to the accumulation of a large number of free radicals within the film, accelerating the photo-oxidative degradation of the film and shortening its functional lifespan to a range of typically 45 to 60 days, which does not satisfactorily meet the requirements for the film’s functionality during crop growth [31,32]. Furthermore, the melting blending of PBAT with PLA and PPC is the mainstream research direction for enhancing biodegradable mulch films. However, due to the structural differences among these three, their compatibility is relatively poor, leading to the inability to form structurally homogeneous degradation film materials, thus impacting their synergistic effects.

To address the challenges mentioned above and enhance the mechanical properties and water vapor barrier performance of PBAT biodegradable mulch films and prolong the degradation period, this research has formulated a composite biodegradable mulch film using PBAT as the primary raw material. The study modified PBAT through polymer blending, using melt blending to prepare modified biodegradable mulch films with PBAT, PLA, and PPC, and added lactic acid–isopropyl carbonate copolymer (PTLA) as a compatibilizer to fuse them, enhancing their overall performance efficiently. As an excellent compatibilizer for PLA and PPC, PTLA enables the fusion of small PLA and PPC aggregates within the large PBAT aggregates, stabilizing the overall structure and reducing the water vapor permeance of the composite biodegradable mulch film, thereby improving its mechanical properties and extending the degradation period.

## 2. Material and Methods

### 2.1. Materials

The polybutylene adipate-co-terephthalate (PBAT) was from Shanxi Jinhui Zhaolong High-tech Co., Ltd. (Xiaoyi, China), the polypropylene carbonate (PPC) was from Jiangsu Zhongke Jinlong Environmental Protection New Materials Co., Ltd. (Taixing, China), and the polylactic acid (PLA) was from Anhui Fengyuan company (Bengbu, China).

### 2.2. Synthesis and Characterization of PTLA

The PTLA copolymer is synthesized through a polymerization process (Figure 1). A combination of biodegradable resins (PLA and PPC) is blended in the ratio of m:n = 4:6 within a stainless steel reactor equipped with a mechanical stirrer and a nitrogen flow device. The mixture is stirred for over 7 h at 175 °C until complete dissolution of PLA and PPC, resulting in a viscous and uniform solution. After the mixture is purged with nitrogen, the reactor is heated to 260 °C and maintained for 2 h under a pressure of 0.2 MPa. Subsequently, the pressure gradually decreases for approximately 1.5 h, followed by the addition of a specific quantity of potassium persulfate (KPS) as the polymerization initiator. The mixture is stirred for 2 h at 200 °C to yield the PTLA copolymer. ^1^H NMR (600 MHz, CDCl_3_): δ 5.23–5.13 (m, 2H), 4.28–4.20 (m, 3H), 2.03 (q, J = 6.4 Hz, 1H), and 1.60–1.49 (m, 15H) (Figure 2).

### 2.3. Preparation of PBAT/PLA-PPC-PTLA Membranes

Prior to the blending process, PBAT, PPC, PLA, and PTLA underwent a drying procedure at 85 °C for 24 h. Subsequently, these materials were combined in varying proportions of 90/5/5/1, 90/5/5/2, 90/5/5/5, 90/5/5/10, 90/5/5/15, and 90/5/5/20 (*w/w/w/w*) to achieve a homogeneous blend, which was then subjected to melt blending using a twin-screw extruder (Labtech Engineering Co. Ltd. (Samutprakarn, Thailand), L/D ratio of 30, Screw Speed range 0–150). The temperature profile ranged from 170 °C to 185 °C, spanning from the feed zone to the discharge port, accompanied by feed and screw speeds of 25 rpm and 170 rpm, respectively. Subsequent to the melt blending process, the materials were cooled using an air-cooled conveyor belt, followed by palletization to obtain PBAT/PPC-PLA-PTLA blended pellets. These pellets were subsequently processed through a single-screw extrusion film blowing machine to produce PBAT/PPC-PLA-PTLA blended thin films with a film blowing temperature set at 185 °C. The resulting product was identified as the PBAT/PPC-PLA-PTLA mulch film, with specific film numbers and formulations documented in Table 1.

### 2.4. Characterization of the Copolymer PTLA

(1)The liquid-state ^1^H NMR spectrum was acquired using a Bruker 600 MHz AVANCE III instrument (Bruker Corporation, Billerica, MA, USA), with TMS as the internal standard.(2)The sample’s infrared spectrum in the wavenumber range of 4000 cm^−1^ to 600 cm^−1^ was obtained using a Fourier transform infrared spectrometer in ATR mode, with a resolution of 4 cm^−1^.(3)The sample’s thermal decomposition was monitored under an N_2_ atmosphere using a thermal gravimetric analyzer (TGA). The temperature range was from 25 °C to 800 °C, with a heating rate of 10 °C/min.

### 2.5. Characterization of PBAT/PPC-PLA-PTLA Blend Mulch

(1)Water Vapor permeance: The water vapor permeance of the samples was tested in accordance with the requirements of GBT1037-1988 using a water vapor permeance tester. Circular samples with an area of 33.2 cm^2^ were cut from the mulch film using a sampler [33]. The samples were placed in the sample chamber of the water vapor permeance tester along with an appropriate amount of ultrapure water. The system output pressure was set to 4–5 MPa, and the pressure of the automatic drying filter was set to 0.3–0.35 MPa. The test temperature and relative humidity were set to 38 °C and 90%, respectively. Each sample was tested twice for repeatability.(2)Mechanical Properties: The longitudinal tensile properties and longitudinal tear resistance properties of the samples were tested using a universal tensile testing machine in accordance with the requirements of GB/T1040.1-2018 and QB/T1130-1991 [34,35]. In the tensile strength test, the film was cut into 10 mm wide and 130 mm long rectangular samples with a gauge length of 100 mm and a test speed of 200 mm/min. In the tear resistance test, the film was cut into samples that met the standard requirements using a mold with a gauge length of 100 mm and a test speed of 200 mm/min. Each sample was tested five times for repeatability, and the average value was taken as the final result.(3)Scanning Electron Microscopy (SEM): The surface microstructure of the modified film was examined using SEM (Thermo Fisher Scientific, ShangHai, China) at an accelerating voltage of 20 kV. Prior to examination, the samples were coated with gold. Before inspection, the coated film was cut into strips, frozen in liquid nitrogen at −80 °C for 1 h, then quickly broken, and the cross-section was used for SEM testing.(4)Fourier Transform Infrared Spectroscopy (FTIR): The Fourier transform infrared spectra of the samples in the wavenumber range of 4000–6000 cm^−1^ were recorded using an FTIR spectrometer (Thermo Fisher Scientific, ShangHai, China), with a resolution of 4 cm^−1^ and testing mode set as ATR mode.

### 2.6. Degradation Behavior of Field Mulch

Surface Degradation Behavior: The apparent degradation of the field mulch was analyzed by visually observing and recording the degree of degradation at fixed observation points. Each observation point covered an area of 50 cm × 50 cm, and the observation height was 5 cm. The degree of mulch degradation was graded according to the following criteria: Grade 0: no cracks present; Grade 1: initial appearance of cracks; Grade 2: small cracks present in 25% of the field; Grade 3: cracks of 2–2.5 cm in the mulch; Grade 4: uniform mesh-like cracks present, no large pieces of mulch remain; Grade 5: almost no mulch present on the surface [36].

Microscopic degradation behavior: The molecular weights of PBAT mulch, PBAT/PLA-PPC-PTLA-10 mulch, and PE mulch samples, subjected to different durations of field exposure, were analyzed using gel permeation chromatography (GPC). Tetrahydrofuran (THF) was used as the solvent, with a mobile phase rate of 1 mL/min, an operating temperature of 40 °C, an injection volume of 100 μL, and a narrow-distribution polystyrene (PS) standard set (TOSOH, Tokyo, Japan) serving as the standard sample.

### 2.7. Insulation Performance Test

The soil temperature of the cultivated layer under the mulch cover and bare soil treatment was measured using a temperature data logger. The probe was buried at a depth of 15 cm and data were recorded every 30 min with two repetitions.

### 2.8. Physiological Indexes and Yield of Crops

A field planting experiment of corn crops was carried out in the Shunyi District of Beijing. The experiment encompassed four treatments, including PBAT/PLA-PPC-PTLA-10 film, PBAT film, PE film, and a bare ground control (CK). During each period, random sampling of five points was conducted for each treatment to measure plant height, stem diameter, leaf area, and fresh weight. In the case of dry matter measurement, plant samples were dried at 120 °C to desiccate, and then further dried at 80 °C to constant weight to measure dry mass.

After harvest, three 5 × 5 m^2^ areas were selected from each treatment to measure plant number, ear number, ear weight, row number, and grain number and calculate the yield. Each treatment was repeated three times and expressed in kg·hm^2^.

## 3. Results and Discussion

### 3.1. Infrared and Thermal Performance Analysis of PTLA

Figure 3 depicts the infrared spectrum of PTLA. The absorption peaks corresponding to C-H stretching vibrations on unsaturated carbons are identified at 2942 cm^−1^ and 2996 cm^−1^. Furthermore, an increase in the intensity of the C=O stretching vibration peak at 1749 cm^−1^ suggests the successful coordination of PLA and PPC, leading to an increase in carboxyl groups within the complex. Notably, a novel absorption peak at 1257 cm^−1^ is observed in the PTLA infrared spectrum, potentially representing the characteristic peak of the C-O-C resulting from the reaction between PLA and PPC, thereby signifying the composite structure of PLA and PPC through ether bonds. Consequently, it can be inferred that the synthesis of the PTLA copolymer has been accomplished.

Figure 4 and Table 2 compare the thermal degradation behavior of PPC, PLA, and PTLA. Pure PPC initiates degradation at 196.66 °C as a result of random chain scission. The initial decomposition temperature (T_98%_, the temperature at which 98% of the weight is lost) of pure PLA is 312 °C, and for pure PTLA, it is 229.5 °C. In contrast to PPC, the thermal weight loss curve of PTLA shifts towards higher temperatures, indicating superior thermal stability compared to PPC. The thermal weight loss rate curves of pure PPC, PLA, and PTLA each exhibit a single peak. The peak temperature (T_max_) for PPC is 273 °C, for PLA is 361.33 °C, and for PTLA, it is 311.16 °C. These results demonstrate that the T_max_ of PTLA shifts towards higher temperatures, indicating enhanced thermal stability and higher thermal processing stability of the copolymer PTLA synthesized from PPC and PLA compared to PPC. PTLA is characterized by a high carbonate content, whereas PLA is distinguished by a significant presence of stable intermolecular hydrogen bonds that exhibit resistance to thermal decomposition, thus leading to enhanced thermal stability.

### 3.2. Morphology of PBAT/PLA-PPC-PTLA Biodegradable Mulching Film

Furthermore, the hydroxyl groups of PBAT can form hydrogen bonds with the ester groups in PLA and PPC, which contributes to the structural stability of the composite biodegradable mulch films. The surface and cross-sectional morphologies of the films were also characterized and are shown in Figure 5 and Figure 6.

It is evident that the surface of the PBAT/PLA-PPC film is smooth but exhibits some granulation. The smooth surface indicates successful melt blending of PBAT with PLA and PPC. At the same time, the presence of particles indicates the presence of incompletely melted degraded resin in regions where hydrogen bonds have not yet formed. In order to enhance the film’s strength, PTLA was added as an excellent compatibilizer for PLA and PPC, enabling the fusion of small PLA and PPC aggregates within the large PBAT aggregates, resulting in improved blending and overall structural stability. As depicted in the figures, the addition of PTLA eliminates the presence of pores on the film’s surface. With an increase in PTLA content, the surface of the composite film becomes progressively smoother, with the PBAT/PLA-PPC-PTLA-10 film demonstrating the best effect. PTLA can effectively promote the fusion of degradable resins. However, the molecular rigidity of PTLA is large, and the melt viscosity is much higher than that of ordinary thermoplastic resins, which makes its molding and processing have certain special characteristics, and the fusion of the film will be reduced if too much is added, preventing it from fusing with PBAT and resulting in the appearance of numerous granules. The surface of the PBAT/PLA-PPC film is not as smooth as that of the PBAT/PLA-PPC-PTLA film, and some granules are observed on its surface.

The obtained PBAT/PLA-PPC and PBAT/PLA-PPC-PTLA mulch films were rapidly cooled and shattered with liquid nitrogen for SEM cross-sectional characterization. The cross-section of the PBAT/PLA-PPC film is not very smooth and may exhibit bubbles in the middle. In contrast, no significant interface separation was observed in the PBAT/PLA-PPC-PTLA mulch film, and with an increase in PTLA concentration, the cross-section gradually became smooth, and the bubbles gradually disappeared, with the PBAT/PLA-PPC-PTLA-10 film demonstrating the best effect, indicating good compatibility between PTLA and the PBAT system.

### 3.3. FTIR Analysis

The FTIR spectra of the PBAT/PLA-PPC and PBAT/PLA-PPC-PTLA mulch films are depicted in Figure 7. The peak at 1450 cm^−1^ is a symmetric stretching vibration of the C-C bond of the aromatic ring, mainly from the aromatic esters in PBAT. The characteristic peak at 1740 cm^−1^ is attributed to the asymmetric stretching vibration of all the C=O groups in the ester bond, including the aliphatic and aromatic esters of PBAT, PLA, and PPC. The peaks at 2875 cm^−1^ represent the symmetric stretching of C-H bonds. The absorption peak at 1280 cm^−1^ corresponds to the stretching vibration of the C-O-C group in PBAT. It can be observed that the intensity of the C-O peak in PBAT/PLA-PPC is similar to that in PBAT/PLA-PPC-PTLA.

Similarly, the absorption peak in the FTIR spectrum at 2954 cm^−1^ corresponds to the stretching vibration of the C–H methyl group, which is also enhanced in PBAT/PLA-PPC-PTLA compared to PBAT/PLA-PPC and PBAT. After the addition of PLA, PPC, and PTLA to PBAT, the absorption peak at 2954 cm^−1^ in the FTIR spectrum of PBAT/PLA-PPC-PTLA broadens, likely due to the methyl group vibrations on PLA, PPC, and PTLA [37,38]. Therefore, it can be inferred that PBAT reacts with PLA, PPC, and PTLA, forming new hydrogen bonds in the mulch film.

### 3.4. Film Mechanical Properties and Water Vapor Barrier Properties

#### 3.4.1. Mechanical Properties

Figure 8 illustrates the tensile loads of PBAT, PBAT/PLA-PPC, and PBAT/PLA-PPC-PTLA films with varying PTLA contents. Mechanical properties are important indicators for biodegradable film testing; the tensile load of the PBAT film is 2.32 N, while that of film No. 1 is approximately 2.78 N. As the PTLA content increases in the PBAT/PLA-PPC-PTLA films, their mechanical performance gradually improves, with the PBAT/PLA-PPC-PTLA-10 film exhibiting the highest tensile load of 3.68 N.

The tear resistance load increases from 2.32 N for the PBAT film to 2.81 N for the PBAT/PLA-PPC film. With an increase in PTLA content, the tear resistance load of PBAT/PLA-PPC-PTLA films also gradually increases, with the PBAT/PLA-PPC-PTLA-10 film exhibiting the highest tear resistance load of 3.48 N. However, with an increase in PTLA content, the tensile load and tear resistance load of the PBAT/PLA-PPC-PTLA films gradually decrease when the PTLA content is above 15%, possibly due to excessive PTLA leading to incomplete compatibility with PBAT, thereby reducing the mechanical performance of the PBAT/PLA-PPC-PTLA films.

#### 3.4.2. Water Vapor Barrier Properties

The water vapor transmission rate can reflect the water vapor barrier performance of the film; the lower the water vapor transmission rate, the lower the number of water molecules between the film through the film, reducing the loss of the film and prolonging the use of time. As shown in Figure 8C, the water vapor permeance of the PBAT film is 702.35 g/(m^2^·24 h). The addition of PPC and PLA reduces the water vapor permeance of the PBAT/PLA-PPC film by 30.24%. The water vapor barrier properties of PBAT/PLA-PPC-PTLA films were ranked in descending order according to the amount of PTLA added: 10%, 5%, 2%, 1%, 15%, and 20%, with the PBAT/PLA-PPC-PTLA-10 film demonstrating the best water vapor performance at 412.34 g/(m^2^·24 h). The addition of PTLA improves the compatibility between PBAT and PPC due to the excellent barrier properties of PPC, enhancing the water vapor barrier performance of the blended films.

### 3.5. Performance Test of Field Mulch

#### 3.5.1. Insulation Performance

Figure 9 depicts a comparison of soil temperatures under different mulch covers. The soil temperature is reported to be higher under the mulch with a superior water vapor barrier performance, attributed to the high heat capacity of water. Throughout the mulching period, the lowest soil temperature is observed for the bare ground control treatment, while the highest temperature is recorded under the polyethylene film cover. In the initial 0–25 days of mulching, the soil temperature under the PBAT/PLA-PPC-PTLA-10 film cover is similar to that under the PBAT film cover, due to the intact state and excellent heat retention properties of both films during this period. Subsequently, from day 25 to 65 of mulching, the soil temperature under the PBAT/PLA-PPC-PTLA-10 film cover surpasses that under the PBAT film cover, possibly due to varying degrees of degradation of the two biodegradable mulch films. The PBAT film exhibits cracks and holes earliest, while the PBAT/PLA-PPC-PTLA-10 film shows them at a later stage. Moving to days 65 to 80 of mulching, the soil temperature under the PBAT/PLA-PPC-PTLA-10 film cover aligns with that under the PBAT film cover, as both films have reached the advanced stage of degradation, leading to reduced water retention and insulation effect. The accumulated soil temperatures within 0–75 days of mulching for the CK treatment, PBAT film cover, PBAT/PLA-PPC-PTLA-10 film cover, and PE film cover are 2069.15 °C, 2167.59 °C, 2252.36 °C, and 2344.81 °C, respectively. The accumulated soil temperature under the PBAT/PLA-PPC-PTLA-10 film cover is 3.9% higher than that under the PBAT film cover and 8.8% higher than that of the bare soil, indicating the superior warming performance of the PBAT/PLA-PPC-PTLA-10 film for soil compared to the PBAT film.

#### 3.5.2. Mechanical Properties

The tensile loads and water vapor permeance of the field mulch films are compared in Figure 10A. With an increase in the number of days of mulching, the tensile loads of the PBAT/PLA-PPC-PTLA-10 film, PBAT film, and PE film show a declining trend, while the tensile load of the PE film fluctuates between 2.5 N and 2.53 N. After 45 days of mulching, the tensile load of the PBAT film decreased from 1.12 N to 1.01 N, a reduction of 10.8%; the tensile load of the PBAT/PLA-PPC-PTLA-10 film decreased from 2.086 N to 2.072 N, a reduction of 3.7%. After 45 days of mulching, the tensile load of the PBAT/PLA-PPC-PTLA-10 film was 73.18% higher than that of the PBAT film. This indicates that during the field degradation process, the mechanical performance of the PBAT/PLA-PPC-PTLA-10 film consistently surpasses that of the PBAT film, with a smaller decrease in mechanical performance, which is beneficial for maintaining the water vapor barrier performance of the mulch films.

#### 3.5.3. Water Vapor Barrier Properties

The water vapor permeance (WVP) of field mulch films increases with the number of days of mulching, with the WVP of the PBAT/PLA-PPC-PTLA-10 film, PBAT film, and PE film showing an increasing trend (Figure 10B). After 30 days of mulching, the WVP of the PBAT film increased from 752.681 g/(m^2^·24 h) to 962.184 g/(m^2^·24 h). The initial water vapor permeance of the PBAT/PLA-PPC-PTLA-10 film was 33.7% lower than that of the PBAT film. This indicates that during the field degradation process, the water vapor barrier performance of the PBAT/PLA-PPC-PTLA-10 film consistently surpasses that of the PBAT film. These results demonstrate the significant potential of the PBAT/PLA-PPC-PTLA film for practical applications in various agricultural contexts.

### 3.6. Degradation Behavior of Field Mulch

#### 3.6.1. Surface Degradation of Mulch

Figure 11 and Table 3 present a comparison of the observable degradation of the PBAT/PLA-PPC-PTLA-10 film, PBAT film, and PE film over 0–75 days of film laying. Additionally, we include fixed-point observation images of the PBAT film at intervals of 0, 15, 30, 45, 60, and 75 days of film laying. It was observed that degradation of the PBAT film commenced after 30 days of film laying, with the emergence of tiny cracks at 45 days, followed by larger cracks at 60 days. Moreover, the fixed-point observation images of the PBAT/PLA-PPC-PTLA-10 film revealed that degradation of the PBAT/PLA-PPC-PTLA-10 film initiated after 60 days of film laying, with tiny cracks appearing after 70 days and larger cracks after 75 days. Finally, the fixed-point observation images of PE film indicated no significant observable degradation over the 0–75-day period of film laying. These findings suggest that the PE film displayed negligible observable degradation during field film laying, and the apparent degradation rate of the PBAT/PLA-PPC-PTLA-10 film is slower than that of the PBAT film, thereby allowing for a longer coverage period.

#### 3.6.2. Microscopic Degradation of Mulch Films

Figure 12 illustrates the molecular weight distribution curve of the field film. At the same time, Table 4 presents the average molecular weight of the field film. In Figure 12, after 15 days, a slight leftward shift in the molecular weight distribution curve indicates a small amount of PBAT degradation. Moreover, at 45 days, a substantial leftward shift of the molecular mass serious chain breakage and degradation of PBAT, resulting in the production of inorganic small molecular substances. At 60 days, the rightward shift in the molecular weight distribution curve signifies the cross-linking of degradation products to form large molecular substances.

Analysis of Figure 12 reveals that the molecular weight of the PBAT/PLA-PPC-PTLA-10 film is concentrated around 86,173 on the day of laying. From 15 to 30 days, there is a slight leftward shift in the molecular weight distribution curve and the appearance of a large molecular substance with cross-linking of degradation products. At 45 days, a significant leftward shift indicates severe chain breakage and degradation. At 60 days, the rightward shift in the molecular weight distribution curve and the increase in the amount of large molecular substances indicate the cross-linking of degradation products.

Table 4 shows that with increasing laying time, the number average molecular weight (Mn) and weight average molecular weight (Mw) of both the PBAT film and PBAT/PLA-PPC-PTLA-10 film decrease, reflecting the degradation behavior of the films in the field. The polydispersity index (Mn/Mw) of these films increases, indicating a wider molecular weight distribution and a mixture of long and short polymer chains. The weight average molecular weight of the PBAT film and PBAT/PLA-PPC-PTLA-10 film laid on the 60th day decreased by 68.37% and 62.43%, respectively, compared to the film laid on day 0. Previous studies have suggested that the reduction in polymer molecular weight decreases the flexibility of molecular chains, increases their mobility, and makes them more susceptible to the instantaneous passage of water vapor, thereby reducing the water vapor barrier properties of the film. Therefore, the water vapor barrier performance of the PBAT film is inferior to that of the PBAT/PLA-PPC-PTLA-10 film due to the greater decrease in molecular weight [39,40].

### 3.7. Physiological Indexes and Yield of Crops

Table 5 presents a comparison of maize yield and its constituent elements under various field mulching treatments. The results indicate that the maize yield was highest in the PBAT/PLA-PPC-PTLA-10 treatment, followed by PBAT, and lowest in the control (CK) treatment. The mulch-covered maize in the PBAT/PLA-PPC-PTLA-10 treatment exhibited significantly higher maize ear length, ear diameter, ear weight, and yield compared to the bare ground treatment. Specifically, the maize ear length of the PBAT/PLA-PPC-PTLA-10 mulch-covered maize was 16.18% higher than that of the CK treatment, the ear diameter was 18.44% higher than that of the CK treatment, the ear weight was 39.89% higher than that of the CK treatment, and the total yield was 40.1% higher than that of the CK treatment. Furthermore, compared to the PBAT mulch cover, the maize ear length, ear diameter, ear weight, and yield of the PBAT/PLA-PPC-PTLA-10 mulch-covered maize increased significantly. The ear length of the PBAT/PLA-PPC-PTLA-10 mulch-covered maize was 7.4% higher than that of the PBAT treatment, the ear diameter was 5.1% higher than that of the PBAT treatment, the ear weight was 5.33% higher than that of the PBAT treatment, and the total yield was 5.45% higher than that of the PBAT treatment. However, there were no significant differences in these parameters between the PBAT/PLA-PPC-PTLA-10 and PE mulch treatments. These findings suggest that the PBAT/PLA-PPC-PTLA-10 mulch cover enhances maize yield, possibly due to its superior water vapor barrier performance, which creates more favorable environmental conditions for maize growth and development. Additionally, the PBAT/PLA-PPC-PTLA-10 mulch provides sufficient water and heat conditions for early-stage crop growth, leading to rapid growth and nutrient accumulation. Even after mulch degradation, the crop can continue to grow normally.

## 4. Conclusions

The combination of PTLA with PBAT, PLA, and PPC materials has resulted in the development of a new fully biodegradable agricultural film, PBAT/PLA-PPC-PTLA, which has been thoroughly analyzed. Experimental findings demonstrate that the incorporation of PTLA improves the compatibility among PBAT, PLA, and PPC, leading to an increase in tensile strength to 3.68 N and a reduction in water vapor permeability to 412.34 g/m^2^·24 h, representing enhancements of 58.62% and 70.33% over pure PBAT film, respectively. Furthermore, the degradation period has been prolonged to 70 days, 10–25 days longer than the PBAT film. Field trials involving the cultivation of corn crops in the Shunyi District, Beijing, have indicated that the new agricultural film, compared to PBAT film, elevates soil accumulated temperature by 3.9% and crop yield by 5.45%, bringing it closer to polyolefin films. As a biodegradable agricultural film, the modified PBAT/PLA-PPC-PTLA film can undergo decomposition after crop harvest, playing a pivotal role in mitigating white pollution. This underscores its substantial potential for sustainable environmental development. Prospective research efforts could further refine the composite film formulation, assess the performance of even more superior biodegradable agricultural films in various crop plantings, and furnish more robust theoretical and experimental substantiation for their regional and crop suitability in practical production.

## Figures and Tables

**Figure 1 polymers-16-01295-f001:**
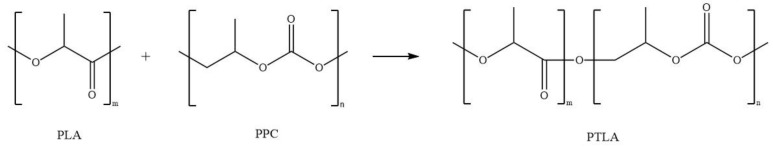
Synthesis route of copolymer PTLA.

**Figure 2 polymers-16-01295-f002:**
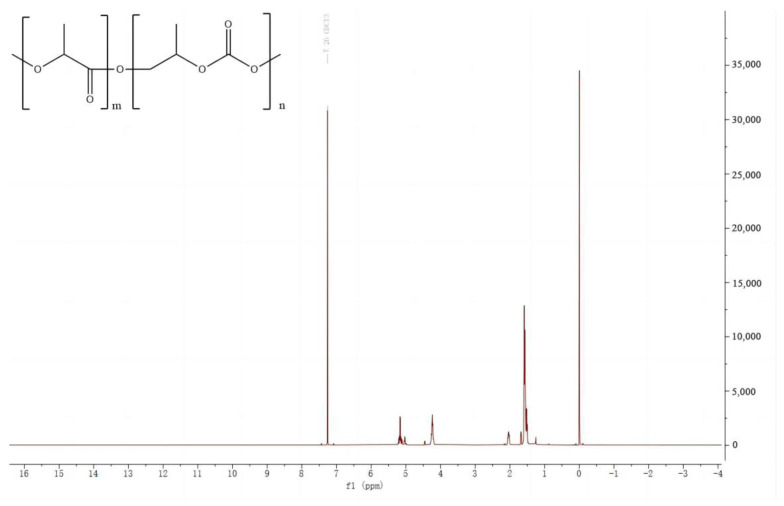
^1^H NMR spectra of the copolymer PTLA.

**Figure 3 polymers-16-01295-f003:**
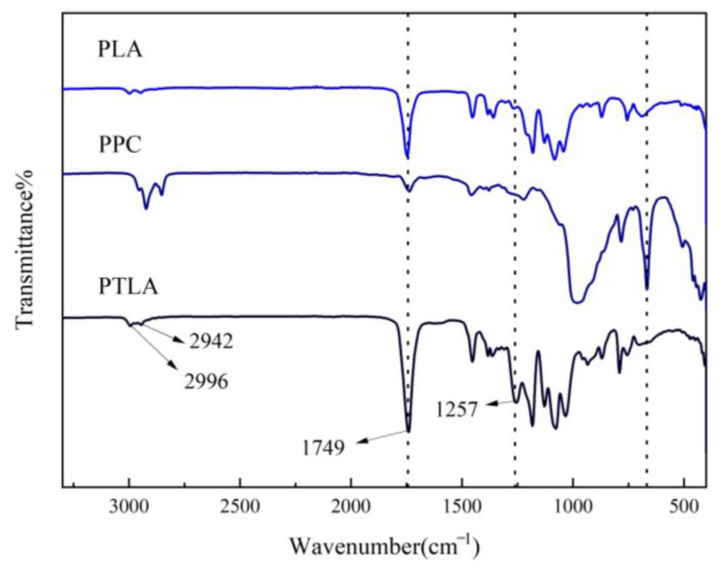
FTIR of the copolymer PTLA.

**Figure 4 polymers-16-01295-f004:**
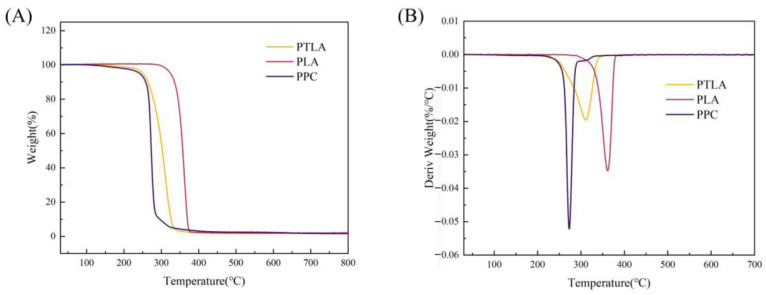
Thermal weight loss curve and thermal weight loss rate curve of copolymer PTLA. (**A**) Weight loss curve; (**B**) weight loss rate curve.

**Figure 5 polymers-16-01295-f005:**
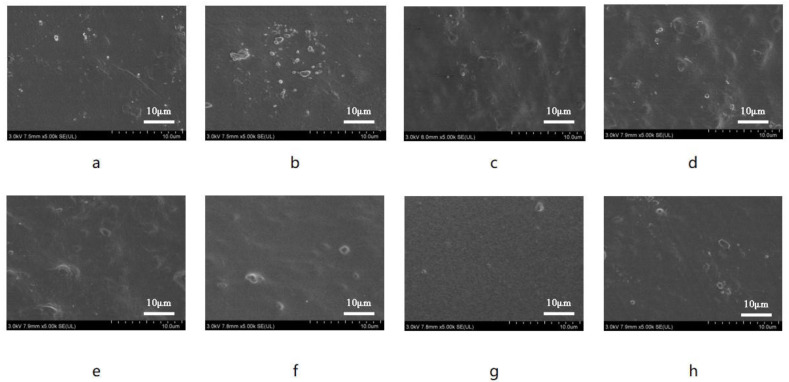
SEM images of the (**a**) PBAT film, (**b**) PBAT/PLA-PPC film, (**c**) PBAT/PLA-PPC-PTLA-1 film, (**d**) PBAT/PLA-PPC-PTLA-2 film, (**e**) PBAT/PLA-PPC-PTLA-5 film, (**f**) PBAT/PLA-PPC-PTLA-10 film, (**g**) PBAT/PLA-PPC-PTLA-15 film, and (**h**) PBAT/PLA-PPC-PTLA-20 film.

**Figure 6 polymers-16-01295-f006:**
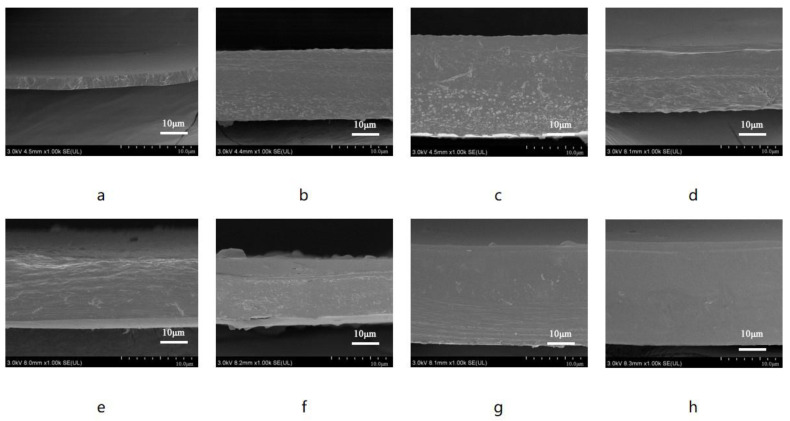
SEM cross-sectional images of the (**a**) PBAT film, (**b**) PBAT/PLA-PPC film, (**c**) PBAT/PLA-PPC-PTLA-1 film, (**d**) PBAT/PLA-PPC-PTLA-2 film, (**e**) PBAT/PLA-PPC-PTLA-5 film, (**f**) PBAT/PLA-PPC-PTLA-10 film, (**g**) PBAT/PLA-PPC-PTLA-15 film, and (**h**) PBAT/PLA-PPC-PTLA-20 film.

**Figure 7 polymers-16-01295-f007:**
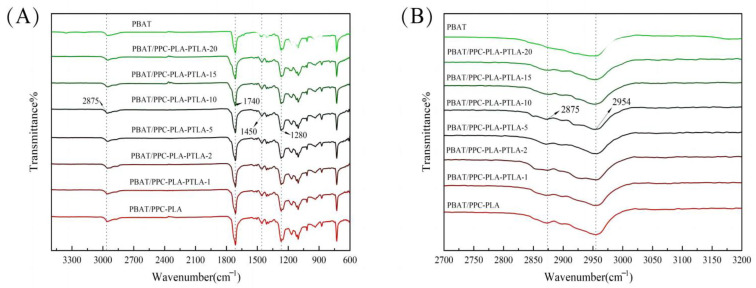
FTIR of different composite biodegradable mulches. (**A**) Wavenumber in the range of 600–3500; (**B**) wavenumber in the range of 2700–3200.

**Figure 8 polymers-16-01295-f008:**
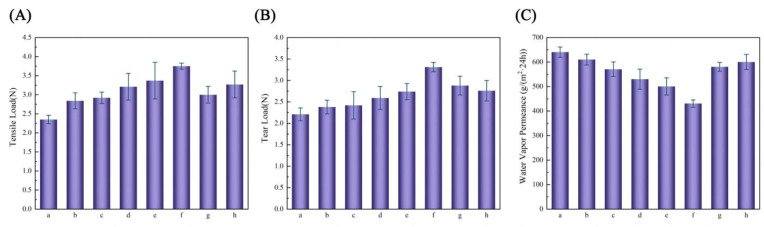
(**A**) Tensile load, (**B**) tear load, (**C**) water vapor permeance of the (a) PBAT film, (b) PBAT/PLA-PPC film, (c) PBAT/PLA-PPC-PTLA-1 film, (d) PBAT/PLA-PPC-PTLA-2 film, (e) PBAT/PLA-PPC-PTLA-25 film, (f) PBAT/PLA-PPC-PTLA-10 film, (g) PBAT/PLA-PPC-PTLA-15 film, and (h) PBAT/PLA-PPC-PTLA-20 film.

**Figure 9 polymers-16-01295-f009:**
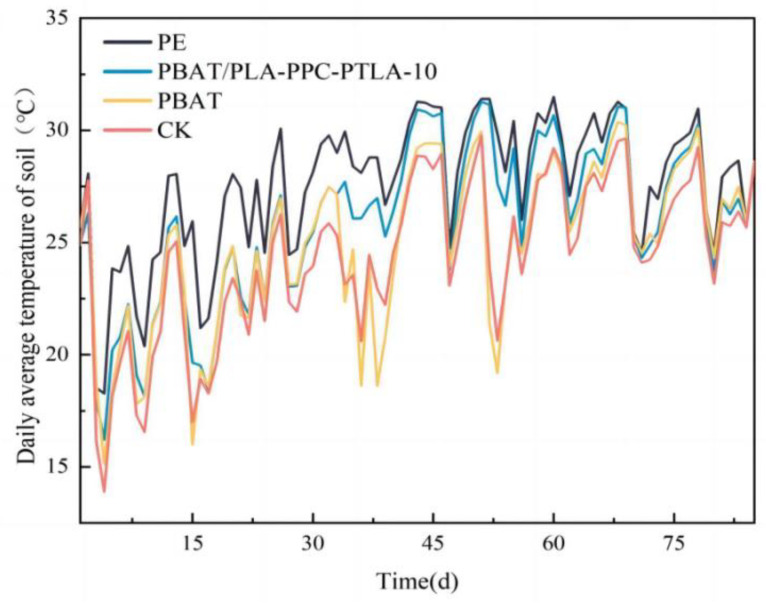
Soil temperature under mulch cover.

**Figure 10 polymers-16-01295-f010:**
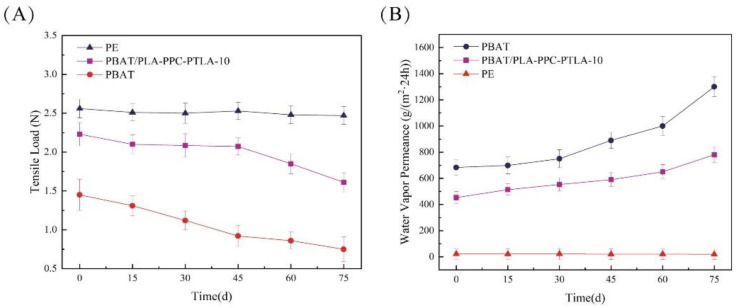
(**A**) Tensile load and (**B**)water vapor permeance of the field mulch films.

**Figure 11 polymers-16-01295-f011:**
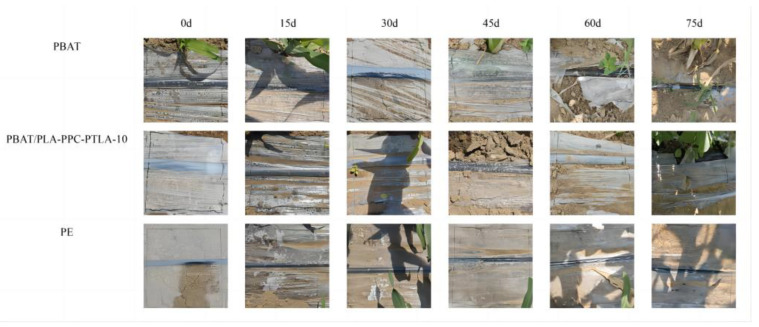
Surface degradation behavior of the PBAT film, PBAT/PLA-PPC-PTLA-10 film, and PE film.

**Figure 12 polymers-16-01295-f012:**
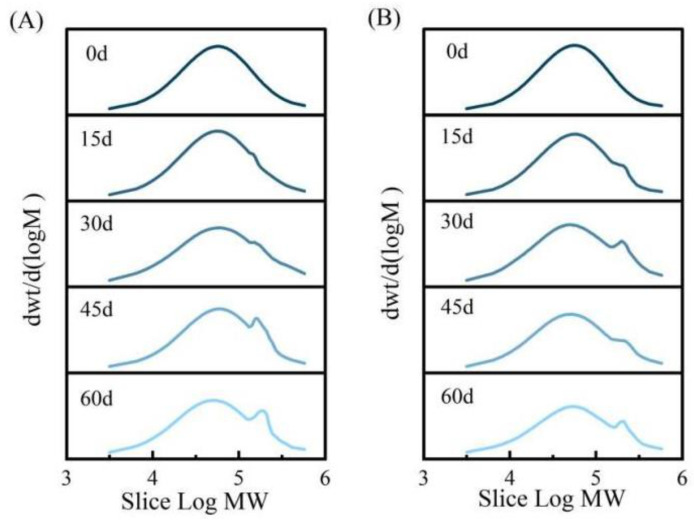
Molecular weight distribution curves of the (**A**) PBAT/PLA-PPC-PTLA-10 film and (**B**) PBAT film.

**Table 1 polymers-16-01295-t001:** Numbering of PBAT/PPC-PLA-PTLA degradation films and their formulation ratios.

Mulch Number	PBAT Contents	PPC Contents	PLA Contents	PTLA Contents
PBAT/PPC-PLA-PTLA	90%	5%	5%	0%
PBAT/PPC-PLA-PTLA-1	90%	5%	5%	1%
PBAT/PPC-PLA-PTLA-2	90%	5%	5%	2%
PBAT/PPC-PLA-PTLA-5	90%	5%	5%	5%
PBAT/PPC-PLA-PTLA-10	90%	5%	5%	10%
PBAT/PPC-PLA-PTLA-15	90%	5%	5%	15%
PBAT/PPC-PLA-PTLA-20	90%	5%	5%	20%

**Table 2 polymers-16-01295-t002:** Thermal degradation behavior of the copolymer PTLA.

Type	T_98%_ (°C)	T_max_ (°C)
PPC	196.66	273
PLA	312	361.33
PTLA	229.5	311.16

**Table 3 polymers-16-01295-t003:** The degradation grade of PBAT/PLA-PPC-PTLA-10 films, PBAT films, and PE films in the field.

Time (d)	0	10	15	25	30	40	45	55	60	70	75
PBAT/PLA-PPC-PTLA-10	0	0	0	0	0	0	1	1	2	2	2
PBAT	0	0	0	0	1	2	2	3	3	3	3
PE	0	0	0	0	0	0	0	0	0	0	0

0: No cracks appear; 1: cracks begin to appear; 2: small cracks appear in 25% of films; 3: 2–2.5 cm cracks appear in films.

**Table 4 polymers-16-01295-t004:** Average molecular weight of field films.

Treatment	Time (d)	M_n_ (Da)	M_w_ (Da)	M_w_/M_n_ (d)
PBAT/PLA-PPC-PTLA-10	0	35,288	86,173	2.462091
	15	34,713	79,606	2.307771
	30	19,541	54,002	2.331982
	45	25,699	71,161	2.583166
	60	7269	32,362	3.573779
PBAT	0	36,194	89,047	2.661893
	15	29,978	79,442	2.851349
	30	12,774	32,620	2.273647
	45	22,973	77,013	3.474457
	60	6721	28,157	4.189406

**Table 5 polymers-16-01295-t005:** Physiological Indexes and Yield of Crops.

Treatment	Ear Length (cm)	Ear Diameter (cm)	Ear Weight (g)	Yield (kg·ha^−1^)
PBAT/PPC-PLA-PTLA-10	20.12 ± 1.23	5.79 ± 0.36	348.15 ± 10.56	24,372.82 ± 218.67
PBAT	18.71 ± 0.81	5.51 ± 0.28	330.51 ± 12.86	23,112.93 ± 261.24
PE	20.43 ± 0.53	5.83 ± 0.15	357.57 ± 6.24	24,994.75 ± 187.69
CK	17.32 ± 0.97	4.89 ± 0.43	248.86 ± 15.98	17,395.81 ± 279.34

Note: The data in the table are average ± standard deviation.

## Data Availability

The raw data supporting the conclusions of this article will be made available by the authors on request.

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
