# Peer review of "Modified Composite Biodegradable Mulch for Crop Growth and Sustainable Agriculture"

_polymers, 2024, doi:10.3390/polym16091295_

Round 1

Reviewer 1 Report

Comments and Suggestions for Authors

Modified Composite Biodegradable Mulch for Crop Growth and Sustainable Agriculture is the result of long research of the authors. There are a lot of tests for film structure study, Physical properties and practical application in real condition. The best content of film was chosen and there properties are compare with analogs. While the article provides some interesting fundamental results, but some details could be improved. Please, for the next submition number each lines.

1. Design of the article are not classical. There are the main parts of the research article, but Material and methods section should be follow after Introduction section. Here, Experimental Section are given after results and discussion. Please, rename ,,Experimental Section,, into ,, Material and methods,, and put it after Introduction section.

2. Introduction section should include the goal of the study.

3. The fist paragraph of 2.1 section is not result or discussion. It’s a protocol of Synthesis and it should be reput into Material and methods section. The last sentences of this paragraph should be explain in more detail.

4. Fig. 1. Is it the only one possible structure of polymer? Is it possible to produces dimers of PLA-PLA or PPC-PPC?

5. Fig. 2 should be discussed in more detail. How this result connected with copolymer structure.

6. What references were used to analysis the IR spectra (fig. 3). Please, add kind of vibration for novel absorption peak at 1257 cm-1. Why on the fig. 3 pointed absorption peak at about 600 cm-1 and are not discussed?

7. Could you explain results of thermal degradation analysis why thermal stability of PLA is higher than PTLA?

8. Table 4 should be added errors bar for all measured temperature and added information how many times was repeated tests.

9. Fig. 5 and 6. Images are very small could you provide here two or three main discussed figs and others put into support information. What the number is mean (PBAT/PLA-PPC-PTLA-10 film)? What is differences between PBAT/PLA-PPC-PTLA-10 film and PBAT/PLA-PPC-PTLA-20 film?

10. Please support the following statement ,,the presence of granules suggests areas where hydrogen bonds,, with relevant research. Why hydrogen bonds are not appear in PBAT/PLA-PPC-PTLA-10? What kind of process are occur during film formation?

11. Please support the following statement ,,large number of carbonate groups in PTLA can make the surface of the composite film excessively adhesive,, with relevant research. Why do carbonate groups effect on adhesive properties?

12. Compare fig. 5 and 6. Why did bubbles gradually disappear after PBAT/PLA-PPC-PTLA-10 composition (fig. 6), but on the fig. 5 there is the minimum bubbles for this composition?

13. Please, add kind of vibration for absorption peak at 1450 cm-1 and 1740 cm-1 (section 2.3).

14. For this statement ,,Therefore, it can be inferred that PBAT reacts with PLA, PPC, and PTLA, forming new hydrogen bonds in the mulch film,, could you support it with relevant reports and provide possible scheme of this process.

15. Section 2.4.1. Could you elaborate this statement ,,Mechanical properties are important indicators for biodegradable film testing,, How Mechanical properties (tensile load and tear resistance load) indicate biodegradable properties of films?

16. Fig. 8. Could you add information how did error bars be estimated? How many tests was carried out?

17. Section 2.4.2. What did Water Vapor Barrier Properties indicate? Why minimum of this parameter is better for Mulch forming? Could you rework this statement ,,The water vapor barrier properties of PBAT/PLA-PPC-PTLA films with different PTLA contents are ranked from high to low as follows: 1-10, 1-5, 1-2, 1-1, 1-15, 1-20,, It’s not clarify this ranking. Could you explain possible reason of water vapor parameter increasing for PBAT/PLA-PPC-PTLA-15 and PBAT/PLA-PPC-PTLA-20 films.

18. Section 2.5.1. Please provide full name for this abbreviators ,, CK treatment,, and ,, PE film,, Please double check average temperatures in the text. Is it 2069.147 ℃, 2167.528 ℃, 2252.357 ℃, and 2344.805 ℃? Approximate this value to a significant figure, taking into account the measurement error.

19. Fig. 11. Images are very small could you provide here two or three main discussed figs and others put into support information. Small cracks are discussed but it’s hard to identify on this small pics.

20. Table 2. There is a mistake in the name, not ,, BAT films,, but ,, PBAT films,,.

The manuscript are provide interesting results and can be published in a journal after some corrections and suggested improvements.

Reviewer 2 Report

Comments and Suggestions for Authors

This paper reports on producing biodegradable mulch films using a ternary blend system (PBAT/PLA-PPC-PTLA). The authors report an improvement of mechanical properties, enhanced water vapor barrier properties, and an extension of the functional period compared to pure PBAT. The manuscript is overall well-written and contains interesting results. However, the originality of this research remains low since the mulch films from the PBAT/PLA-PPC system have already been reported. 

For instance, Wang et al. Polymer Bulletin (2023) 80:2485–2501 reported a similar mulch film production system (without polymerization of PLA-PPC).

In my opinion, the paper is worth to be published in Polymers after a minor revision.

My comments:

- Please indicate the specific ratio of PLA-PPC during the polymerization process (Fig 1). 

-  In '3.2 Preparation of PBAT/PLA-PPC-PTLA membranes' 

Please indicate the extruder type (manufacturer, L/D, total feed rate).  

Comments on the Quality of English Language

It is a well-written manuscript with a few mistyping and mistakes. Please check it again.

Author Response

For research article

Response to Reviewer 2 Comments

1. Summary

Thank you very much for your kind consideration and meticulous work on our manuscript. We deeply appreciate your constructive and professional comments and your affirmation on our work will greatly encourage us. The comments are very helpful for revising and improving our paper. According to your suggestions, we make a careful revision on our manuscript and carried out additional supporting experiments. The revised portions are marked in red in the paper and the main corrections are listed as following. 

2. Questions for General Evaluation

Reviewer’s Evaluation

Response and Revisions

Does the introduction provide sufficient background and include all relevant references?

Yes

Are all the cited references relevant to the research?

Yes

Is the research design appropriate?

Yes

Are the methods adequately described?

Yes

Are the results clearly presented?

Yes

Are the conclusions supported by the results?

Yes

3. Point-by-point response to Comments and Suggestions for Authors

Comments 1: [Please indicate the specific ratio of PLA-PPC during the polymerization process (Fig 1).]

Response 1: [ Thank you for your professional suggestion.We changed the specific ratio in the article, on page 2, fifth paragraph, fifth line. ] 

Comments 2: [In '3.2 Preparation of PBAT/PLA-PPC-PTLA membranes' .Please indicate the extruder type (manufacturer, L/D, total feed rate).]

Response 2: [Thank you for your professional suggestion.We changed the device parameter (twin-screw extruder(Labtech Engineering Co. Ltd,L/D ratio of 30,Screw Speed range 0-150)) in the article, on page 3, first paragraph, fourth line. ]

4. Response to Comments on the Quality of English Language

Point 1:It is a well-written manuscript with a few mistyping and mistakes. Please check it again.

Response 1:Yes.Thanks for your input.

5. Additional clarifications

[We changed Boru Dong's email [email protected] to [email protected].]
